# Comparative Analysis of Comprehensive Genomic Profile in Thymomas and Recurrent Thymomas Reveals Potentially Actionable Mutations for Target Therapies

**DOI:** 10.3390/ijms25179560

**Published:** 2024-09-03

**Authors:** Filippo Lococo, Elisa De Paolis, Jessica Evangelista, Andrea Dell’Amore, Diana Giannarelli, Marco Chiappetta, Annalisa Campanella, Carolina Sassorossi, Alessandra Cancellieri, Fiorella Calabrese, Alessandra Conca, Emanuele Vita, Angelo Minucci, Emilio Bria, Angelo Castello, Andrea Urbani, Federico Rea, Stefano Margaritora, Giovanni Scambia

**Affiliations:** 1Thoracic Surgery, Università Cattolica del Sacro Cuore, 00168 Rome, Italy; jessica.evangelista@policlinicogemelli.it (J.E.); stefano.margaritora@policlinicogemelli.it (S.M.); 2Thoracic Surgery, Fondazione Policlinico Universitario A. Gemelli IRCCS, 00168 Rome, Italy; marco.chiappetta@policlinicogemelli.it (M.C.); annalisa.campanella@guest.policlinicogemelli.it (A.C.); carolina.sassorossi@guest.policlinicogemelli.it (C.S.); 3Clinical Chemistry, Biochemistry and Molecular Biology Operations (UOC), Fondazione Policlinico Universitario A. Gemelli IRCCS, 00168 Rome, Italy; elisa.depaolis@policlinicogemelli.it (E.D.P.); andrea.urbani@policlinicogemelli.it (A.U.); 4Departmental Unit of Molecular and Genomic Diagnostics, Genomics Core Facility, Gemelli Science and Technology Park (G-STeP), Fondazione Policlinico Universitario A. Gemelli IRCCS, 00168 Rome, Italy; 5Thoracic Surgery Unit, Department of Cardiac, Thoracic, Vascular Sciences and Public Health, University of Padova, 35122 Padova, Italy; andrea.dellamore@aopd.veneto.it (A.D.); federico.rea@unipd.it (F.R.); 6Epidemiology and Biostatistics Facility, Gemelli Science and Technology Park (G-STeP), Fondazione Policlinico Universitario A. Gemelli IRCCS, 00168 Rome, Italy; diana.giannarelli@policlinicogemelli.it; 7Unit of Pathology, Fondazione Policlinico Gemelli IRCCS, 00168 Rome, Italy; alessandra.cancellieri@policlinicogemelli.it; 8Pathology Unit, Department of Cardiac, Thoracic, Vascular Sciences and Public Health, University of Padova, 35122 Padova, Italy; fiorella.calabrese@unipd.it (F.C.); alessandra.conca@guest.policlinicogemelli.it (A.C.); angelo.minucci@policlinicogemelli.it (A.M.); 9UOSD Oncologia Toraco-Polmonare, Comprehensive Cancer Center, Fondazione Policlinico Universitario Ago-stino Gemelli IRCCS, 00168 Rome, Italy; emanuele.vita@guest.policlinicogemelli.it (E.V.); emilio.bria@policlinicogemelli.it (E.B.); 10UOC Oncologia Medica, Ospedale Isola Tiberina—Gemelli Isola, 00186 Roma, Italy; 11Nuclear Medicine Department, Fondazione IRCCS Ca’ Granda Ospedale Maggiore Policlinico, 20122 Milan, Italy; angelo.castello@policlinico.mi.it; 12Division of Gynecologic Oncology, Fondazione Policlinico Universitario A. Gemelli-IRCCS, 00168 Rome, Italy; giovanni.scambia@policlinicogemelli.it

**Keywords:** NGS, recurrent thymoma, surgery, CGP, gene profile, clinical trials

## Abstract

Molecular profiles of thymomas and recurrent thymomas are far from being defined. Herein, we report an analysis of a comprehensive genetic profile (CGP) in a highly selected cohort of recurrent thymomas. Among a cohort of 426 thymomas, the tissue was available in 23 recurrent tumors for matching the biomolecular results obtained from primary and relapse samples. A control group composed of non-recurrent thymoma patients was selected through a propensity score match analysis. CGP was performed using the NGS Tru-SightOncology assay to evaluate TMB, MSI, and molecular alterations in 523 genes. CGP does not differ when comparing initial tumor with tumor relapse. A significantly higher frequency of cell cycle control genes alterations (100.0% vs. 57.1%, *p* = 0.022) is detected in patients with early recurrence (<32 months) compared to late recurrent cases. The CGPs were similar in recurrent thymomas and non-recurrent thymomas. Finally, based on NGS results, an off-label treatment or clinical trial could be potentially proposed in >50% of cases (oncogenic Tier-IIC variants). In conclusion, CGPs do not substantially differ between initial tumor vs. tumor recurrence and recurrent thymomas vs. non-recurrent thymomas. Cell cycle control gene alterations are associated with an early recurrence after thymectomy. Multiple target therapies are potentially available by performing a comprehensive CGP, suggesting that a precision medicine approach on these patients could be further explored.

## 1. Introduction

Thymomas are relatively rare tumors of epithelial thymic cells, representing approximately 0.2–1.5% of all malignancies [1]. From a pathological point of view, the World Health Organization classifies them into different types (A, AB, B1, B2, and B3) based on the ratio between non-tumoral lymphocytic components and the resemblance to normal thymic architecture [2].

Despite these tumors presenting usually with an indolent behavior, the natural history is often unpredictable, with recurrences reported in 10–30% of patients even 10 to 20 years [3,4] after radical resection (R0).

Tumor recurrences are generally located in the chest (mostly in the pleural cavity) and are usually treated with loco-regional approach (combined or not with systemic treatment), with surgery remaining the gold standard approach when technically feasible [4].

Indeed, several studies [5,6,7] and meta-analyses [8] have reported improved early and long-term outcomes after surgery in recurrent thymoma patients, whereas few authors support chemotherapy only (usually platinum-based protocols) in this setting [9,10].

Unfortunately, the clinical history of these tumors is very insidious: indeed, even after re-do surgery, further recurrences of disease are very frequently reported [4,7]; similarly, in recurrent cases that underwent first-line treatment, disease progression is quite common and further lines of therapy are not standardized and become generally much less effective. Consequently, there is an urgent need for novel treatments for recurrent and platinum-resistant thymomas.

In the last decade, the wide implementation of high throughput technologies and comprehensive genomic profiling (CGP) in solid tumors allowed the identification of a broad spectrum of molecular aberrations and altered signaling pathways in thymic epithelial tumors (TET), leading to the definition of distinct molecular profiles in TETs. Several attempts to identify somatic mutations that characterize TETs have been made in recent years. Target-specific drugs for TETs have not been developed, because genomic aberrations in TETs are poorly understood [11].

Several studies have generally explored thymomas and thymic carcinoma in the same dataset [11,12], clearly demonstrating different biological aspects in terms of tumor mutational burden (TMB), microsatellite instability (MSI) status, and molecular pathways. In particular, TMB has been reported to be much higher in thymic carcinoma [11,12,13], and this stays as a predictive factor of immune check point inhibitors (ICI) efficacy. A recent meta-analysis [14] suggested that ICI could be a therapeutic option for selected patients with thymic carcinoma who are not feasible to obtain a curative radical treatment after first-line chemotherapy. On the other hand, no impressive changes in therapeutic paradigm of unresectable/recurrent thymomas that progressed to platinum-based chemotherapy have been achieved so far.

In this framework, we reviewed a large cohort of surgically resected thymomas, performing CGPs on the surgical specimen of both primary and recurrent thymomas. A control group of non-recurrent thymomas was also selected (propensity score match analysis) and their gene profiles also analyzed. The final aims of the present study are:−To compare the CGP of recurrent thymoma versus non-recurrent thymoma patients;−To explore the CGP of both primary and recurrent thymomas and identify associations with clinicopathological variables;−To evaluate actionable mutations detected in thymomas as targets for new therapeutic approaches.

## 2. Results

### 2.1. Clinical and Pathological Characteristics

The main patients’ characteristics and pathological features of Rec-_Thy Group and NoRec_Thy are summarized in Table 1. In detail, the Rec-_Thy patients were quite young (median age = 51 yrs.) and presented a recurrence several months after thymectomy (median disease-free interval (DFI) of 32 months). They had mostly Masaoka Stage II–III Type-B thymoma and were often treated with neoadjuvant therapy before surgery. These variables were balanced in the control group (NoRec_Thy) with similar distribution of age, Masaoka Stage, and histology (see Table 1). According to the classification reported above, thymomas were classified at high-risk in 75% of Rec_Thy and 64.3% of NoRec-_Thy.

### 2.2. Overall Genomic Results (Entire Cohort)

Globally, the majority of patients showed reportable oncogenic/likely oncogenic molecular alterations, observed in 81% of cases, with a low rate of oncogenic mutations/case (0–6) and with several genes appearing only once in the cohort (Figure 1 and Appendix A).

Recurrent defective pathways were identified. Molecular alterations observed in genes involved in the cell cycle were recurrent in this study, with amplifications in the CCND3 (16%), CDK4/6 (27%), and MDM4 (32%) genes being the most frequent.

Alterations in DNA damage repair (DDR) pathways including homologous recombination (HR), nucleotide excision repair (NER), and mismatch repair (MMR) were identified. In particular, loss-of-function (LoF) SNVs in DDR were identified in 22% patients, without any recurrently mutated gene. Among HR, we identified LoF mutations in BRCA1, RAD51C, RAD54L, and CHEK2. One patient resulted as carrier of MLH1 LoF oncogenic mutation, with a predicted impairment of the MMR system.

Other dysregulated pathways included RTK family signaling (with FGFR1/4 amplifications in 5% of patients) and PI3K/AKT/mTOR activation (ESR1 and PIK3CA genes in 5% of patients).

Amplifications in the MYC oncoprotein family (MYC, MYCL, MYCN genes) were identified in 10% of patients. In addition, alterations in epigenetic regulatory genes as TET2 and DNMT3A were rarely identified in the cohort (8% of patients). TP53 oncogenic variant was identified in one case in our series (3%).

TMB status resulted low across all samples, together with MSI stable status (i.e., MMR-proficient). Only for one patient we observed a high TMB, probably related to MLH1 mutation, which could lead to the accumulation of somatic frameshift and SNVs. For this patient, we were not able to calculate the MSI status (failure to cover the 130 MSI sites). According to ESMO guidelines, follow-up-germline testing was not recommended for the enrolled patients.

### 2.3. CGP Differences in Recurrent Thymoma vs. Non-Recurrent Thymoma

From the comparative evaluation of the two study groups of Rec_Thy and NoRec_Thy, no overall significant differences emerged in the molecular analysis (Table 2). Oncogenic alterations were reported in 83% of recurrent thymomas vs. 78% of non-recurrent thymomas (*p* = 0.76). The rate of clinically relevant alterations (Tier-IIC) is similar in the two groups, with 43% of recurrent thymomas vs. 57% of non-recurrent thymomas. Looking at the distribution and types of oncogenic alterations, the same percentage of cases with dysregulation of the two main pathways of cell cycle (74% in recurrent vs. 64% in non-recurrent) and DDR (22% in recurrent vs. 21% in non-recurrent) was identified. Even though observed in a limited number of cases, dysregulations in epigenetic regulatory genes and PI3K/AKT pathway genes were identified only in the non-recurrent group of thymomas (14%). On the other hand, alterations in RTK–RAS family signaling cascade were detected only in recurrent thymomas (FGFR1/4, BRAF) (13%). No differences in the MSI and TMB status were identified between the two groups.

### 2.4. CGP Differences in Primary vs. Recurrent Thymoma and Inter-Relationship with Clinic-Pathological Variables

No significant differences in CGP emerged from the comparative evaluation of matched primary and recurrence tissue biopsies. As reported in Table 3, similar frequencies of samples with at least one oncogenic/likely oncogenic alteration were observed when comparing primary thymomas and recurrent thymomas (Kappa statistics −0.049 *p* = 0.84; McNemar *p* = 0.73).

In detail, genes belonging to the cell cycle pathway were similarly altered in both primary (37%) and their recurrences (50%) (Kappa statistics −0.09 *p* = 0.30, McNemar *p* = 0.69). Comparable results were obtained evaluating the distribution of DNA damage repair alterations, occurring at 19% of primary tumor and 12% at matched recurrences (Kappa statistics 0.29 *p* = 0.23, McNemar *p* = 0.99). TMB was low in both primary thymomas and their recurrences, with no remarkable changes among samples.

On the other hand, when evaluating the distribution (see Table 4) of at least one genomic alteration in Rec_Thy with early recurrence (DFI < 32 months), we found a higher proportion of samples with at least one mutation compared to Rec_Thy with DFI > 32 months (100% vs. 71.4%, *p* = 0.082).

Interestingly, more cell cycle control gene alterations were observed in early recurrence Rec_Thy when compared with others (100.0% vs. 57.1%, *p* = 0.022), while a similar distribution of alteration of gene of DNA repair (25% vs. 25%, *p* = 0.99) was found.

Finally, exploring the associations between other clinical variables and gene mutations (identified in any type of specimen in this group, i.e., primary and/or recurrence), we observed a significantly higher frequency of genetic alteration in DNA repair pathways in early Masaoka Stage tumors (see Table 4). A similar gene profile distribution was found according to age, presence of M.G., histology, and classes of risk.

### 2.5. Actionable Mutations for New Therapeutic Approaches

Overall, based on CGP profiling, off-label treatments approved in different disease entities or clinical trials potentially recruiting patients with mutated TETs have been identified. No directly actionable genomic alterations (classifiable as Tier I) could be identified in our patients, due to the lack of an FDA/EMA approved molecular-targeted therapy in thymoma clinical setting. Looking into the global actionability, approved treatments or clinical trials could be potentially recommended for 49% analyzed patients (18 out of 37). Appendix A showed clinical trials potentially including thymomas in which the molecular characterization of the tissue sample and the presence of a specific biomarker are enrollment criteria. Approved or experimental therapies mainly encompass cyclin-dependent kinase (CDK) inhibitors, PARP (poly ADP-ribose) inhibitors, and tyrosine kinase (TKs) inhibitors.

## 3. Discussion

In this study, we took advantage of a robust series of recurrent TETs for which a comprehensive GCP was led and compared with a control group of non-recurrent thymomas. Taken together, our results provide a unique insight into the molecular pathways activated in recurrent thymomas, paving the way for precision medicine approaches using targeted agents or experimental drugs in a large part of them. To best of our knowledge, our cohort is the largest reported so far focusing on recurrent thymoma, this representing a specific subset of thymomas for which the standard of care is still a matter of debate.

Despite recent evidences [8] promoting the role of surgical treatment for recurrent thymomas, the high rate of re-recurrences [3,4,5,16] suggests that surgery alone may fail to achieve a complete disease control at this stage. On the other side, systemic treatment including immune check point inhibitors (ICI) [17,18] or somatostatin receptor targeting therapies (alone or with prednisone) [18,19] showed controversial results.

Because of this, the strategy of care in recurrent thymomas is still an intriguing issue of exploring the role of molecular-targeted strategies after/prior to surgical resection.

In the present study, the CGP data confirm a relatively low mutational burden, as emerged from literature [11,12,13,20,21,22]. The majority of studies highlight a limited number of molecular alterations, with no gene found to be mutated with a frequency exceeding 10% [11,12,13,20,21].

This may partially explain the paucity of effective molecular target therapy. The literature data about pre-clinical and clinical evaluation of target drugs in TETs showed attractive results mainly in the thymic carcinoma (TC) context [23].

Looking into the global actionability of our molecular findings, approved treatments or clinical trials could be potentially recommended for almost 49% of thymoma patients herein analyzed. Similar data emerged from the EORTC-SPECTA/Arcagen study for rare tumors (53.8%) [20] and a lower percentage (27%) from the SPECTRALung platform [21].

The recommendations mainly encompassed CDK inhibitors, PARPi, RTK inhibitors, and PI3K/mTOR inhibitors. Loss of cell cycle control emerged as a common occurrence in thymomas [21,23,24] and the most recurrent in our cohort (27%). Targeting D-type cyclins in tumors expressing amplified CDK4/6 and CCND3 is widely investigated in solid and hematological malignancies (see Appendix A). A growing number of CDK inhibitors are currently tested in clinical trials enrolling advanced/recurrent solid tumors as pan-CDK inhibitors or more selective CDK inhibitors.

Palbociclib, Ribociclib, and Abemaciclib are FDA-approved for hormone receptor-positive (HR+) breast cancer treatment. For patients with TETs, the utility of Palbociclib and Milciclib maleate CDK inhibitors (PHA-848125AC) are under investigation in the phase II NCT03219554 and NCT01301391 trials, respectively. Pre-clinical and phase I supporting studies highlighted that in thymomas no expression of p21 and p27 (natural inhibitors of CDKs) significantly correlates with poor prognosis for disease-free survival [25] and objective partial response type B3 and C thymic malignancies [26].

Interestingly, in the present CGP analysis we found a significant higher alteration rate of cyclin group genes in patients who experienced an early recurrence if compared with others (100.0% vs. 57.1%, *p* = 0.022), this suggesting a potential link between these genes and the biological aggressiveness in thymomas.

Moreover, CDK4/6 pathway hyper-activation is associated with worse prognosis in TC [27]. It is known that many other proteins interact with CDK4/6 and modulate the cell cycle, such as MDM2/MDM4 and TP53. TP53 mutation has been reported in approximately 3% of thymomas, as also identified in the present study [23,28]. MDM4 is significantly amplified (14% up to 43%) in several cancer types [29]. Here we identified MDM4 alterations in a similar percentage (32%).

Additionally, DDR pathway alteration was reported in the 22% of patients. We did not identify recurrent mutated targets in this subset. Defects in HRD pathway represent the molecular basis of the synthetic lethality of PARP inhibition, and FDA/EMA approved drugs are available in different settings (Olaparib, Talazoparib, and Rucaparib). The role of DDR was largely unexplored in TETs. Few literature observations, mainly BRCA1/2 and ATM, are available about single case or families with sporadic/recurrent thymomas [30,31,32]. Among these, a patient with BRCA2-mutated thymoma showed a significant clinical benefit from treatment with Olaparib, with imaging showing overall stabilization of her disease [33].

Recommendations also encompassed TK inhibitors. Experimental and clinical data suggested the potential role of VEGFR1/3 and FGFR1/4-driven angiogenesis dysregulation in TETs. [34]. The pan-RTK inhibitor Sunitinib is currently in NCCN guidelines for the treatment of advanced TC, and is under investigation in a phase II clinical trial enrolling TC and thymoma patients [35]. Additionally, the NCT02307500 clinical trial evaluating the multikinase inhibitor Regorafenib is active for thymoma B2/3 patients in progression after chemotherapy.

Finally, alterations in PI3K/AKT/mTOR pathway are present in 5% of the cases in our series, according to the TGCA PanCancer Atlas. Pre-clinical data suggested that this subset of thymomas activates the PI3K pathway through the up-regulation of a large microRNA cluster on chr19q13.42 with a marked reduction of cell viability [36]. In this context, the insulin-like growth factor-1 receptor (IGF1R) inhibitors cixutumumab and the mTOR inhibitor everolimus were investigated, with a partial response (NCT00965250 and NCT02049047, respectively) [37,38]. Everolimus is in the NCCN guidelines for the treatment of thymomas and TC in progression after chemotherapy. The modest activity of buparlisib, an oral pan-PI3K inhibitor, in relapsed or refractory thymomas, resulted from the NCT02220855 clinical trial [39]. These studies provide evidence to support further evaluation of PI3K/Akt pathway targeting in patients with advanced thymoma.

Mutations in epigenetic regulatory genes as DNMT3A was reported in ~7% of thymomas in the AACR GENIE cohort (https://genie.cbioportal.org/ accessed on 21 June 2024) and in 10% of TETs in the literature (together with TET gene alterations) [33]. From our data, three patients from the No_Rec groups resulted as carriers of DNMT3A and TET2 alterations.

Finally, no remarkable changes in the molecular profile emerged from the comparative analysis of recurrence vs. non-recurrence thymoma groups and from the in-patient analysis of primary vs. recurrence samples. These finding suggest a molecular stability over time and a low grade of tumoral heterogeneity in terms of clonal evolution, all supported by the low TMB that seems to be not affected by the tumor progression. To note, the evaluation of tumor heterogeneity and evolution includes and integrates molecular, cellular, and architectural variability aspects [40].

Probably, a more in-depth analysis using a multi-omics approach is needed to better grasp these features in TETs. At the same time, the overlap between primary tumors and metastases may justify the use of CGP at the time of recurrences instead of retrospectively evaluating primary lesions.

At present, it is quite hard to integrate refractory or recurrent TETs into large phase clinical trials, mainly due to the rarity of the disease. Advantages in CGP adoption also relies in the possibility to access large clinical trial designed to adopt the best target therapy according to genetic alterations (e.g., CUSTOM trial NCT01306045, NCT05667948, and NCT01385722).

### Limitations, Points of Strength and Future Clinical Applications

This study presents some limitations, concerning both the selection of cases and the methodology of the analysis. First, thymoma patients have been selected in a relatively long interval (>20 years) in a retrospective study. However, thymomas are quite rare tumors and recurrences are uncommon and usually occurring several years after surgery. Thus, a long observation time may be necessary to have an acceptable number of cases and enough follow-up to perform survival analyses. However,, although this is to our knowledge the study with the largest number of recurrent thymoma patients with GCP analysis, the sample size limited the generalization of our results, which would need more cases to be confirmed. Concerning the methodology of analysis, FFPE specimens aged > 10 years from the present analysis, results in a lower DNA sample quality or sequencing failures (NGS metrics quality) (see Figure 1). It has to be noted that we adopted one of the largest CGP panels available to allow a wide molecular investigation. Although robust, the assay has some limitations at the time of writing. The assay validation for copy loss was lacking and the variant calling analysis for indels was restricted to ≤25 bp of length, which precludes reporting potentially deleterious alterations. Additionally, due to the specific analytical characteristics of the sequencing solution adopted, we cannot replicate some of the TET molecular data previously available in the literature (e.g., copy number loss of CDKN2A/B genes, absence of GTF2I gene in the panel etc.).

At the same time, we would like to enhance the points of strength of the present study. First, the novelty of the topic analyzed in the study is that it is an emerging and unexplored issue for future research and clinical applications. Indeed, since we showed up to 50% of our recurrent thymoma patients presented with TIER II-C molecular alterations, these results are opening some opportunities for innovative molecular-targeted strategies in this setting.

Moreover, considering that prospective studies on thymomas are clearly not feasible, the adoption of a comparative group of NoRec_Thy selected from a propensity score match analysis from our real-world institutional cohort of patients clearly stays as an added value.

Finally, the bioinformatic and analytical analysis of sequencing findings as here described represent a state-of-the-art approach for clinical translational studies. The NGS panel used contains a comprehensive pool of genes clearly associated with tumor biological characterization and with clinical relevance in terms of target therapy and trial enrollment. This feature maximized the interpretation of genomic results for translational purposes, allowing a proper integration with clinical data.

Concerning the clinical application of this study, while we have clearly showed that a CGP may be of high value for the management of (recurrent) thymomas, we need to consider the remarkable costs related to CGP analysis and its overall clinical usefulness before suggesting the adoption of CGP analysis on a large scale. In this setting, the identification of the best candidates, who would really benefit from CGP, is a crucial point; while performing CGP analysis in all thymoma patients is questionable, considering that only 14% of them will experience a relapse [4,5], we may suggest testing only recurrent cases. Since the gene profile does not change when comparing initial tumor with tumor relapse (as reported herein), CGP analysis may be performed on the initial surgical tissue if the sample of the recurrent tumor is not available.

We strongly believe that in the future, a precision medicine approach will be probably available for these patients only if phase I/II clinical trials will be finally conducted and if promising molecules will be proposed in this setting.

In the present study, we would like to provide the proof of principle to stimulate this process, which today seems to be very challenging.

## 4. Materials and Methods

### 4.1. Study Design and Selection of Cases

This bicentric, observational, retrospective, cohort study was reviewed and approved by the Fondazione Policlinico Universitario “Agostino Gemelli” IRCCS, ethics committee and partner institutional review boards (study identification: 3027). Data on the patients treated for thymoma recurrence from January 1, 2003, to January 1, 2023, in two high-volume centers were collected and retrospectively reviewed. The two centers were selected because they had high-volume, long-term experience and similar management in thymoma and recurrent thymoma patients. This study was conducted on the base of an overall surgical cohort of 426 TETs. Thymic carcinomas and neuroendocrine thymic tumors were excluded from the analysis, because of their biological and molecular differences from thymomas. After excluding cases with missing data, we selected patients who experienced a recurrence during follow-up (see Consort Diagram, Figure 2). A control group was made through propensity score match analysis, being composed of thymoma patients who did not develop any recurrence during a minimum of five years after initial surgery. A total of 43 patients experienced a thymoma recurrence after thymectomy (Overall Relapse Rate: 13.65%). Tissue was available for the analysis in 23 of them, these composing the Recurrent Thymomas Group (Rec_Thy). After seven NGS workflow failures in one sample (primary or recurrent tumor), both primary and relapses tumor were analyzed and compared in 16 cases (see Consort Diagram, Figure 2). Finally, 23 thymoma patients who have not reported any recurrence after at least five years thymectomy were identified from the entire surgical cohort of cases thanks to a propensity score match analysis (as reported below). Tissue was available in 14 cases, these composing the control group of non-recurrent thymoma (NoRec_Thy).

### 4.2. Pathological Review

Thymomas were classified using the Masaoka–Koga staging classification [41], the eighth edition of the TNM staging system for thymic neoplasm [42], and the WHO classification system for TETs [43]. They were also divided into two groups on the basis of a previously reported prediction model of recurrence [15]. According to this model, patients with T1–T2 thymomas or T3 type A–AB–B1 thymomas had a significantly lower incidence of recurrence (“low-risk group”) than those with T2–T3 type B2–B3 thymomas (“high-risk group”). All patients underwent surgery for both primary and recurrent thymoma with a curative aim. We excluded patients with only radiological suspicion of thymoma recurrence or with pathological confirmation achieved by small biopsy. A dedicated pathologist in each involved center (AC, FC) reviewed all the specimens and re-evaluated the histology according to the WHO classification system. A centralized revision at the promoting center was performed in cases with doubts.

### 4.3. Comprehensive Genomic Profiling and Bioinformatics Analysis

Eosin-stained histology tissue slides were examined by dedicated pathologists to identify areas of at least 20% of tumor cell content. DNA was extracted using the AllPrep^®^ DNA/RNA FFPE commercial kit (QI-AGEN^®^, Hilden, Germany), according to manufacturer’s procedures. Nucleic acid quality was assessed by using the Illumina Infinium FFPE QC kit (Illumina^®^, San Diego, CA, USA) on the CFX Connect Real-Time PCR Detection System instrument (Bio-Rad^®^, Hercules, CA, USA). The DNA quantitation was performed using the Qubit HS dsDNA fluorimetric assays (Life Technologies^®^, Waltham, MA, USA) and only samples with a quantity greater than 40 ng were analyzed.

A pancancer CGP was performed using the TruSight Oncology 500 High-Throughput (TSO500HT) assay. TSO500HT allows identification of low-frequency somatic variants as single nucleotide variants (SNVs), insertions and deletions (indels), splice variants, and copy number alterations (CNVs, i.e., gain/amplification) in 523 genes related to cancer susceptibility and treatment, along with the major immunotherapy biomarkers (TMB, MSI) (Appendix A). Genomic DNA was sheared and converted into libraries with the addition of unique molecular identifiers (UMI). The NGS was performed on the NovaSeq6000 platform (Illumina). The output data evaluation was obtained using Velsera Clinical Genomics Workspace tool and only genomic profiling characterized by a sequencing data with a median depth of coverage > 500 X was considered in the final review of molecular results. A cut-off of 5% of Variant Allele Frequency (VAF%) was adopted. MSI was calculated from 130 loci.

Final reporting of the detected variants was performed using the Velsera Clinical Genomics Workspace platform (Pierian). The knowledgebase platform integrates real-word medical interpretations and classifications of users with public data sources. In particular, the biological and clinical interpretation was performed according to public population and mutational repositories (e.g., gnomAD [44] (https://doi.org/10.1038/s41586-023-06045-0 accessed on 21 June 2024), COSMIC [45], OncoKb [46,47], ClinVar [48]), FDA and EMA approved labels, NCCN, ASCO, and ESMO guidelines, trials from clinicaltrials.gov and EUCT, and pertinent literature evidences. The tier classification system of the AMP, ASCO, and CAP was adopted [49]. Only molecular alterations predicted to be oncogenic/likely-oncogenic were evaluated (Tier I-II). r. Molecular alterations were considered clinically relevant (Tier-IIC) if targetable by drugs available in different clinical contexts or matched enrollment criteria in a registered clinical trial for the specific clinical context. Variants of unknown significance (Tier III) were excluded. The ESMO Precision Medicine Working Group recommendations were considered for a follow-up germline target test according to annotations in germline mutational databases, types of alteration, and VAF% [50].

### 4.4. Statistical Analysis

As a first step, a propensity score approach was used to select the control group grom patients without recurrence; thanks to this method, we identified a subgroup of patients without recurrence within five years after surgery to perform CGP on. The propensity score approach was based on the nearest neighbor method, with a caliper of 1.5 standard deviations considering sex, age, presence of myasthenia gravis, Masaoka–Koga staging, and histology.

As the second step, comparing CGP on primary tumors and recurrence on the same patient, a paired approach was implemented: kappa statistics was used to measure concordance in the presence of alterations and the McNemar test was used to assess marginal homogeneity

As an overall approach, data were summarized using absolute counts and percentages for categorical items, and median and range when referring to quantitative variables. Association among different pathways and clinical and demographical characteristics were assessed through the chi-square test. IBM-SPSS v.28.0 and R v.4.1.2 software was used for analysis.

## 5. Conclusions

In the present analysis, we found that relevant molecular findings of recurrent TETs generally belong to the cell cycle control pathway. CGP does not substantially differ between initial tumor versus tumor recurrence and recurrent thymomas versus non-recurrent thymomas. Cell cycle control gene alterations are associated with an early recurrence after thymectomy. Multiple target therapies are potentially available, performing a comprehensive CGP, suggesting that a precision medicine approach on these patients may be further explored.

## Figures and Tables

**Figure 1 ijms-25-09560-f001:**
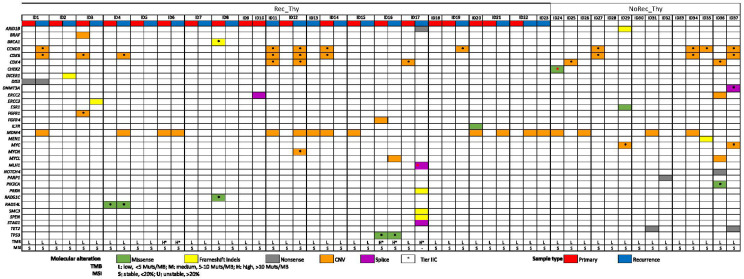
**CGP profile of Thymomas.** The figure shows the molecular alterations and the genomic signatures identified in the cohorts of Rec_Thy (ID1-ID23; primary and recurrence sample types) and NoRec_Thy (ID24-ID37).

**Figure 2 ijms-25-09560-f002:**
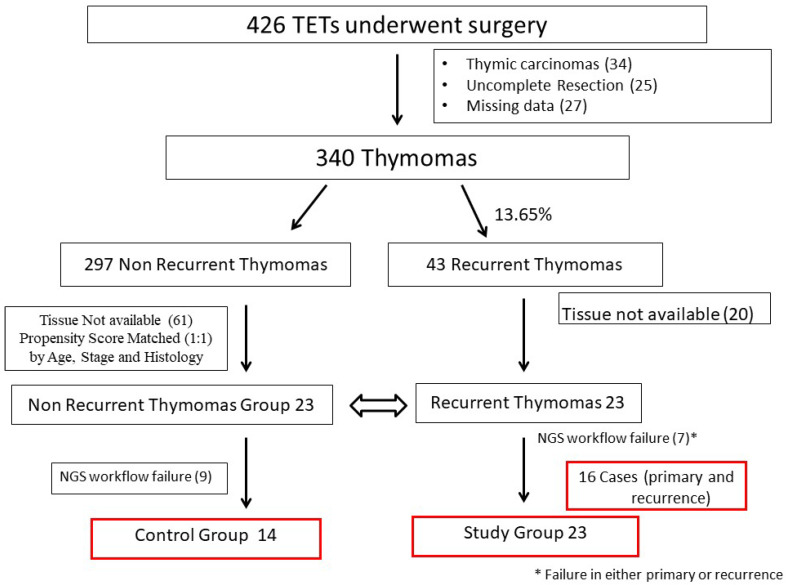
Consort Diagram of the Study Population.

**Table 1 ijms-25-09560-t001:** The main patients’ characteristics and pathological features of Rec-_Thy Group and NoRec_ Thy.

	Rec_Thy (n = 23 pts)	No Rec_Thy (n = 14 pts)
GENDER M F	13 (56.5%) 10 (43.5%)	10 (71.4%) 4 (28.6%)
AGE (median, range)	51 y (27 y–83 y)	59 y (16 y–82 y)
Myasthenia Gravis (MG)	8 (34.8%)	7 (50.0%)
MASAOKA * II III IV	5 (21.8%) 12 (52.2%) 3 (13.0%)	5 (35.7%) 7 (52.2%) 2 (14.3%)
NEOADJUVANT TREATMENT	10/20 (50.0%)	6/14 (42.9%)
HISTOLOGY WHO AB B1 B2 B3	0 (0%) 9 (39.2%) 7 (30.4%) 7 (30.4%)	1 (7.1%) 2 (14.3%) 9 (64.3%) 2 (14.3%)
^ RISK CLASS * Low-Risk High-Risk	5 (25.0%) 15 (75.0%)	5 (35.7%) 9 (64.3%)
STAGE * II III IV	4 (20.0%) 11 (55.0%) 5 (25.0%)	6 (42.8%) 5 (35.7%) 3 (21.5%)
DFI (median, range) **	32 m (6 m–132 m)	/
ADJUVANT TREATMENT	7 (30.0%)	7 (50.0%)

* Only on primary tumor; ** DFI = time between thymectomy and relapse (months); ^ according to criteria reported in [15].

**Table 2 ijms-25-09560-t002:** Distribution of genetic alterations between recurrent thymomas and non-recurrent thymomas.

GROUP	All Patients (#37)	Rec_Thy(#23)	NoRec_Thy (#14)	*p*-Value
Pathway cell cycle	26 (70%)	17 (73.9%)	9 (64.3%)	*p* = 0.53
Pathway DNA repair	8 (22%)	5 (21.7%)	3 (21.4%)	*p* = 0.98
At least 1 alteration	30 (81%)	19 (82.6%)	11 (78.6%)	*p* = 0.76
Clinically relevant alteration	18 (49%)	10 (43%)	8 (57%)	*p* = 0.83

**Table 3 ijms-25-09560-t003:** Distribution of genetic alterations between primary thymomas and recurrent thymomas.

GROUP	Primary_Thy	Recurrent_Thy	*p*-Value
Pathway cell cycle	6 (37.5%)	9 (56.2%)	*p* = 0.30
Pathway DNA repair	2 (12.5%)	3 (18.7%)	*p* = 0.23
At least 1 alteration	9 (56.2%)	11 (68.7%)	*p* = 0.84

**Table 4 ijms-25-09560-t004:** Inter-relationship between clinic-pathological variables and gene mutations in recurrent thymoma (any specimen, see text).

	Pathway Cell Cycle	Pathway DNA Repair	At Least 1 Alteration
Rec_Thy (n = 23)	11/23 (47.8%)	3/23 (13.0%)	14/23 (60.9%)
Masaoka Stage II (n = 5) III-IV (n = 18)	*p* = 0.121 5/5 (100.0%) 12/18 (66.6%)	***p* = 0.019** 3/5 (60.0%) 2/18 (11.1%)	*p* = 0.351 5/5 (100.0%) 14/18 (77.8%)
Age <51 (n = 11) >51 (n = 12)	*p* = 0. 896 5/11 (45.6%) 6/12 (50.0%)	*p* = 0.635 1/11 (9.1%) 2/12 (16.7%)	*p* = 0. 582 8/11 (72.7%) 6/12 (50.0%)
Myasthenia Gravis Yes (n = 8) No (n = 15)	*p* = 0.661 3/8 (37.5%) 8/15 (53.3%)	*p* = 0.960 1/8 (12.5%) 2/15 (13.3%)	*p* = 0.695 4/8 (50.0%) 10/15 (66.7%)
RISK Class * Low (n = 5) High (n = 18)	*p* = 0.121 4/5 (80.0%) 9/18 (50.0%)	*p* = 0.770 1/5 (20.0%) 4/18 (22.2%)	*p* = 0.201 5/5 (100.0%) 11/18 (61.1%)
DFI <32 months (n = 9) >32 months (n = 14)	***p* = 0.022** 9/9 (100.0%) 8/14 (57.1%)	*p* = 0.960 2/9 (22.2%) 3/14 (21.4%)	*p* = 0.082 9/9 (100.0%) 10/14 (71.4%)

* Risk Classes as defined in [15].

## Data Availability

Data are available from the Authors upon request.

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
