# Peer review of "Comparative Analysis of Comprehensive Genomic Profile in Thymomas and Recurrent Thymomas Reveals Potentially Actionable Mutations for Target Therapies"

_ijms, 2024, doi:10.3390/ijms25179560_

Round 1

Reviewer 1 Report

Comments and Suggestions for Authors

Given the rarity of the tumor type and the heterogeneity of data that exists describing the molecular profile of these tumors, there is a need in the field for the type of study described in this paper. While overall, this study is of interest and has value to the field, the presentation of the data significantly limits the utility.

Major comments:

-       Table 1 appears to be a confusing mix of the final group of the non-recurrent thymoma with sequencing data and the entire recurrent thymoma group (not excluding NGS failures). This does not accurately reflect the patients used for the final analysis. Upon review of other tables and data there seems to be 23 cases in the relapse cohort? But this is at odds with Figure 1.

-       Figure 2 needs more explanation. The origin of the variants observed in the recurrent patients is not at all clear. These patients have primary and matched relapse data but what is displayed is not at all clear. Is this a union, only the primary, or only the relapse data? The authors state that the primary and recurrence specimens have a very similar molecular profile but there is no data provided to support this. As one of the novel aspects of this study, this is a major failure.

-       Throughout the manuscript, there are grammatical errors that need to be corrected before publication.

-       The paragraph beginning at Line 387 requires more explanation or removal. Stating that TETs should not be included in clinical trials but celebrating the possible clinical trials that would be relevant for patients in this cohort given the results of this study is confusing.

-       The lack of discussion comparing this study to prior sequencing work is disappointing and blunts any emphasis on novelty. For example, I am surprised there are no CDKN2A/B deletions based on prior reports.

-       The utility of this manuscript to the field would be improved by providing the complete gene and variant list as a supplemental table.

Minor comments:

-       Please define the acronym TET used on line 84 as this has multiple meanings.

-       Line 169 should include references for the “expertly curated genomic databases” as these range widely in their content and citing their use is beneficial to such resources as well as other experts in the field.

-       Given the sequencing failures in the control group, why were additional controls not picked from the set of >200 specimens with tissue available to replace these for a stronger 1:1 match to the recurrence group? Figure 1 only mentions NGS failures but the discussion mentions DNA extraction failures as well. Additional sequencing is costly but for pure DNA extraction failures, these should be possible to replace. Realistically, I understand this is not always feasible but it does reduce the overall impact of the study.

-       The resolution of figure 2 is poor, making it difficult to read.

-       Citations 20-22. COSMIC, ClinVar, and OncoKB all have recommended publications to use for their citations. As publicly available resources, appropriately citing them helps support their continued development and financial backing.

-       Line 289 is missing the * that this statement I believe is referring to.

-       Lines 305, 371, and 396 have a typo in the acronym.

Comments on the Quality of English Language

Grammatical errors throughout. Overall everything is understandable but not polished.

Author Response

Reviewer #1

Major comments:

Comment:      Table 1 appears to be a confusing mix of the final group of the non-recurrent thymoma with sequencing data and the entire recurrent thymoma group (not excluding NGS failures). This does not accurately reflect the patients used for the final analysis. Upon review of other tables and data there seems to be 23 cases in the relapse cohort? But this is at odds with Figure 1.

Reply: Firstly, I would like to thank You for this constructive comment that gives us the opportunity to clarify this point. Actually, as you correctly stated, in the initially submitted manuscript there was a certain confusion between data reported in table 1 and Figures.

Basically, we have two cohort of patients: Study Group with recurrent disease and Control group with no recurrent disease (23 cases, both).  These cohort of patients were obtained by a propensity score analysis. Among study group (n=23 pts) only in a part of cases ( n= 16 pts) we were able to perform a NGS comparative analysis (primary vs recurrence tumor) due to 7 failures in NGS analysis .

Inspired by Your proper comment, we have modified both figure 1 and figure 2 in order to clarify our population to the Readers. In Consort diagram the two populations (Study group and Control Group) are more clearly identificable; similarly, in the other figure we have summarized the overall NGS data of the 23 patients with recurrent thymoma but we have also graphically reported individually NGS data of the primary tumor and NGS data of the recurrence.

In this new version, data reported in Table 1 are in line with those graphically reported in Figures. Moreover, we have clarified the population in the section 4.1. “Study design and selection of cases”

We really thank You for your support in improving the quality of the manuscript.

Correction: see Figures

Comment: Figure 2 needs more explanation. The origin of the variants observed in the recurrent patients is not at all clear. These patients have primary and matched relapse data but what is displayed is not at all clear. Is this a union, only the primary, or only the relapse data? The authors state that the primary and recurrence specimens have a very similar molecular profile but there is no data provided to support this. As one of the novel aspects of this study, this is a major failure.

Reply: We thank the referee for pointing out this aspect and we agree that in the initially submitted version of the manuscript the figure could be cryptic for the readers. As reported above, according to Your suggestion we provided a novel version of the Figure 1 (ex fig. 2) with NGS profiling of primary and recurrence samples showed individually. This also help the Readers to better understand the table 1 (see comment above) that is now in line with data reported in the revised version of Figure 1. We also changed the relative caption.

Correction: see Figure 1

Comment:   Throughout the manuscript, there are grammatical errors that need to be corrected before publication.

Reply: thank you for your observation, language has been revised

Correction: through the text

Comment:   The paragraph beginning at Line 387 requires more explanation or removal. Stating that TETs should not be included in clinical trials but celebrating the possible clinical trials that would be relevant for patients in this cohort given the results of this study is confusing.

Reply: Thanks for such insight comment. Actually, we agree that planning/completing clinical trials on recurrent or refractory thymoma patients is almost hard because of the rarity and this is also the reason why, at today, pharma economy is not so interested in deeply investigating this orphane disease.

However, our statement “…Refractory or recurrent TETs should not be integrated into clinical trials, mainly due to the rarity of the disease…” has too much emphasis, as You correctly noted. Probably is more correct to say:  

“At today, it’s quite hard to integrate refractory or recurrent TETs into large phase clinical trials, mainly due to the rarity of the disease. “

Despite this, we strongly believe that in a next future, a precision medicine approach to this subset of patients will be probably available only if phase I/II clinical trials will be finally conducted and if promising molecules will be proposed in this setting.

In the present study, we’d like to provide the proof of principle to stimulate this process that at today seems to be very challenging.

Correction: We have modified in the text the statement as reported above and we have better explained our opinion in the discussion session.

Comment:   The lack of discussion comparing this study to prior sequencing work is disappointing and blunts any emphasis on novelty. For example, I am surprised there are no CDKN2A/B deletions based on prior reports.

Reply: We thank the referee for this comment that give us the opportunity to clarify this point. As reported in the “Limitations, Points of strength and future clinical applications” section of the paper, we used a large CGP panel that we routinely adopt in clinical context for solid tumor. However, as correctly observed by the You, we cannot replicate some of the molecular results available in literature for TETs due to specific analytical characteristics of the sequencing solution adopted. The TSO500 assay by Illumina is not validated for deletion and, for example, the multigene panel not include the GTF2I gene (see “Copy Number Variant Calling” and “Copy Number VCF” sections of the TruSight Oncology 500 v2.2 Local App User Guide at https://emea.support.illumina.com/downloads/trusight-oncology-500-v2-2-local-app documentation.html). Accordingly to the suggestion, we provided more details in the “Limitations, Points of strength and future clinical applications” section of the revised manuscript and added the full gene content of the test in the supplementary material (Supplementary table 1, section “Overall Genomic Results” page 4).

Comment:  The utility of this manuscript to the field would be improved by providing the complete gene and variant list as a supplemental table.

Reply: According to Your suggestion we provided the full list of the gene content of the NGS assay adopted (Supplementary table 3, section “Comprehensive Genomic Profiling and Bioinformatics Analysis”) and the variant list (Supplementary table 1, section “Overall Genomic Results”).

Minor comments:

Comment:  Please define the acronym TET used on line 84 as this has multiple meanings.

Reply: thank you for your comment. The definition has been added

Correction: see line 83

Comment:  Line 169 should include references for the “expertly curated genomic databases” as these range widely in their content and citing their use is beneficial to such resources as well as other experts in the field.

Reply and correction: We thank the referee for this comment. We revised the manuscript to better clarify the tertiary analysis and the final interpretation. Briefly, we adopted a knowledgebase platform named Velsera Clinical Genomics Workspace by Pierian that integrated several public databases to create a final Tier classification of variants that could be revised by the user. We added more information in the “Comprehensive Genomic Profiling and Bioinformatics Analysis” section of the Materials & Method

Comment:  Given the sequencing failures in the control group, why were additional controls not picked from the set of >200 specimens with tissue available to replace these for a stronger 1:1 match to the recurrence group? Figure 1 only mentions NGS failures but the discussion mentions DNA extraction failures as well. Additional sequencing is costly but for pure DNA extraction failures, these should be possible to replace. Realistically, I understand this is not always feasible but it does reduce the overall impact of the study.

Reply: We have appreciated Your comment and its constructive intent. In the Consort diagram we have clarified that NGS failure was referred to all NGS workflow (both DNA extraction and NGS). Also in the text we have clarified this point.

Since this is a spontaneous no profit study we have had no financial support for NGS analysis a part from our institutional resources. At the beginning, we have planned to analyse all available cases with recurrent thymoma and its control group (23 vs 23, see Figure 2). Unfortunately, the failure rate was almost high and this creating a certain imbalance between groups (see Figure 2).

We have discussed with our statistical expert (D.G.) about the opportunity to replace these cases by recovering them from the entire “surgical cohort” and we tried to do that as following reported.

When performing a new propensity score match analysis, we have to exclude the 23 cases already selected and also patients with tissue not available (n=61), patients with missing  data (n=81) and those (n=91) without a minimum of 5 years of FUP with no missing data (inclusion criteria).  The high number of patient with missing data is justified by the fact that Policlinico Gemelli Hospital (Catholic University of the Sacred Heart) is the biggest hospital in Italy and the surgical landmark for the South part of Italy. Thus several patients were addressed to our surgical department and then followed in other peripherally located hospital.

Therefore,  with the final aim of achieving a 1:1 match between cases of the Study Group and cases of the Control Group, we have tried to perform on the remaining 64 patients a further propensity score approach (as done before) based on the nearest neighbor method with a caliper of 1.5 standard deviations considering sex, age, presence of Myasthenia Gravis, Masaoka-Koga staging and histology.

Only other 5 cases perfectly matched the characteristics of those included Study group. Considering our NGS failure rate, we have done a prevision to obtain only 3 or 4 cases more to be included in the Control Group. Honestly, we avoided investing our money knowing that we could not achieve our final goal (1:1 matched pairs). The raw database with all data is available for the necessary checks.

Comment:  The resolution of figure 2 is poor, making it difficult to read.

Reply and correction: A new version of the figure with high resolution has been included in the manuscript and a pdf version has been also updated.

Comment:  Citations 20-22. COSMIC, ClinVar, and OncoKB all have recommended publications to use for their citations. As publicly available resources, appropriately citing them helps support their continued development and financial backing.

Reply and correction: Thank you for your comment. The citations have been corrected, as suggested (novel reference number: 43, 44, 45, 46, 47)

Comment:  Line 289 is missing the * that this statement I believe is referring to.

Reply: thank you for this comment. Correction has been added

Correction: see table 4

Comment: Lines 305, 371, and 396 have a typo in the acronym.

Reply: thank you for this observation, correction have been made

Correction: see line 222, 288 e 312

Reviewer 2 Report

Comments and Suggestions for Authors

Results presented in the manuscript are original and relevant with potential application in managing patients with recurrent thymomas. Data regarding genomic profile of thymomas have been published previously, but the value of this manuscript is on focusing on comparison of genetic profiles in recurrent vs. non-recurrent thymomas and comparing cancer samples from the same patients before and after recurrence. The group of patients analysed is relatively small, but since recurrent thymomas are rare, the results are valuable.

The methods are appropriate and well described and presentation of the results in the manuscript is clear and adequate.

The discussion if completely focused on potential therapeutic targets emerging from the genomic alterations found in this study which is relevant. However, you should discuss in more details results regarding comparison of recurrent vs. non-recurrent thymomas and comparison of the samples from the same patients of primary and recurrent lesions. So I suggest to include this in the discussion, particularly in the context of cancer inter-lesional heterogeneity and cancer clonal evolution.

There are several minor corrections in the manuscript that should be made:

1. you should explain abbreviation TET the first time it is used (line 84)

2. in figure 1 don't use symbol #, just number is enough

3. in table 1 and in text (line 285) you should explain abbreviation MG

4. lines 212-213, min and max is redundant, just numbers "(0-6)" is enough

5. line 325, use full words instead of abbreviation TC

6. line 341, negative expression is not clear, I suggest no expression

7. line 385, "should not" is misleading, you should rephrase it to emphasize that it is complicated to organize clinical trials with this patients due to small number

8. line 247, which two main pathways?, please explain

Comments on the Quality of English Language

The manuscript is generally readable and understandable but still there are many instances of incorrect wording and syntax so I suggest to proofread the whole manuscript

Several examples are:

line 63, 'distinguish' (better word would be 'classify')

line 63, 'so-called' is redundant

line 66, 'present' instead of 'presented'

line 68, only one 'after' is enough

line 78, '1st' instead of '1rst'

line 92, 'was reported' instead of 'has reported'

line 97, 'that' progressed

line 137, 'classified' or 'grouped' instead of 'organized'

line 154, 'quantity' instead of 'quantitation'

line 199, 'relatively' instead of 'relative'

line 218, change 'accounted on' to more appropriate phrase

line 249, change 'accounted' to a more appropriate word

line 253, 'between' instead of 'in'

line 270, change 'modification' to a more appropriate word

line 306, 'provide a unique' instead of 'provide with a unique'

line 316, 'today' instead of 'at today'

line 317,  change 'where' to a more appropriate word

line 343, 'in type...'

line 354, 'we did not identify' instead of 'we not identified'

line 365, change 'assumed' to a more appropriate word

line 369, 'chemotherapy' instead of 'chemiotherapy'

lines 392, 399, and 426 'almost' is redundant

line 406, 'the novelty of the topic analysed in the study is that it is an...'

line 409, 'results are opening'

Also, there are several instances of inappropriately split words with hyphens (ob-served, thy-momas, be-cause,...)

Author Response

Reviewer #2

Comment: Results presented in the manuscript are original and relevant with potential application in managing patients with recurrent thymomas. Data regarding genomic profile of thymomas have been published previously, but the value of this manuscript is on focusing on comparison of genetic profiles in recurrent vs. non-recurrent thymomas and comparing cancer samples from the same patients before and after recurrence. The group of patients analysed is relatively small, but since recurrent thymomas are rare, the results are valuable. The methods are appropriate and well described and presentation of the results in the manuscript is clear and adequate.

Reply: thanks a lot for Your comment on the manuscript

Comment: The discussion if completely focused on potential therapeutic targets emerging from the genomic alterations found in this study which is relevant. However, you should discuss in more details results regarding comparison of recurrent vs. non-recurrent thymomas and comparison of the samples from the same patients of primary and recurrent lesions. So I suggest to include this in the discussion, particularly in the context of cancer inter-lesional heterogeneity and cancer clonal evolution.

Reply: We thank the referee for this comment. The evaluation of tumor heterogeneity and clonal evolution is a major topic and a key aspect of future clinical research (e.g. liquid biopsy, resistance mechanisms etc). Consequently, we agreed with your relevant suggestion. From our analysis no remarkable changes emerged from the evaluation of the two groups of rec vs not rec TETs and also between primary and recurrence biopsies of the same patients. Looking into the low frequency of mutations/sample and the overall low TMB, these findings suggested a molecular stability of TETs over time that is not affected by tumor progression. Probably, to investigate these aspects in “stable” tumors like TETs, is needed to apply a complex and integrated approach. We underlined this aspect in the discussion.

Correction: see Discussion

Comment: There are several minor corrections in the manuscript that should be made:

  1. you should explain abbreviation TET the first time it is used (line 84)

Reply: thank you for your comment. The definition has been added

Correction: see line 85 

  1. in figure 1 don't use symbol #, just number is enough

Reply: thank you for this comment. The symbol has been removed

Correction: see figure 2

  1. in table 1 and in text (line 285) you should explain abbreviation MG

Reply: thank you for your comment. The abbreviation has been corrected

Correction: see table 1

  1. lines 212-213, min and max is redundant, just numbers "(0-6)" is enough

Reply: thank you for your comment. The correction has been made

Correction: see line 130

  1. line 325, use full words instead of abbreviation TC

Reply: thank you for your comment. The abbreviation has been corrected

Correction: see line 244

  1. line 341, negative expression is not clear, I suggest no expression

Reply: thank you for this observation, correction have been made

Correction: see line 261

  1. line 385, "should not" is misleading, you should rephrase it to emphasize that it is complicated to organize clinical trials with this patients due to small number

Reply: thank you for your comment. The sentence has been rephrased

Correction: see line 305 - 306

  1. line 247, which two main pathways?, please explain

Reply: We thank the referee for this comment. As you correctly stated, it was important to specify the pathways, so they have been added in the text

Correction: see line 165-166

Comments on the Quality of English Language

The manuscript is generally readable and understandable but still there are many instances of incorrect wording and syntax so I suggest to proofread the whole manuscript

Several examples are:

line 63, 'distinguish' (better word would be 'classify')

Reply: thank you for this observation, correction have been made

Correction: line 63

line 63, 'so-called' is redundant

Reply: thank you for this observation, correction have been made

Correction: line 63

line 66, 'present' instead of 'presented'

Reply: thank you for this observation, correction have been made

Correction: line 66

line 68, only one 'after' is enough

Reply: thank you for this observation, correction have been made

Correction: line 68

line 78, '1st' instead of '1rst'

Reply: thank you for this observation, correction have been made

Correction: line 78

line 92, 'was reported' instead of 'has reported'

Reply: thank you for this observation. We corrected with “has been reported”

Correction: line 92

line 97, 'that' progressed

Reply: thank you for this observation, correction have been made

Correction: line 97

line 137, 'classified' or 'grouped' instead of 'organized'

Reply: thank you for this observation. We corrected with “divided”

Correction: line 396

line 154, 'quantity' instead of 'quantitation'

Reply: thank you for this observation, correction have been made

Correction: line 413

line 199, 'relatively' instead of 'relative'

Reply: thank you for this observation. We corrected with “quite young”

Correction: line 112

line 218, change 'accounted on' to more appropriate phrase.”

Reply: thank you for this observation, correction have been made

Correction: line 136

line 249, change 'accounted' to a more appropriate word

Reply: thank you for this observation, correction have been made

Correction: line 170

line 253, 'between' instead of 'in'

Reply: thank you for this observation, correction have been made

Correction: line 171

line 270, change 'modification' to a more appropriate word

“TMB was low in both primary thymomas and their recurrences, with no notable changes observed between samples (data not shown)”

Reply: thank you for this observation, correction have been made

Correction: line 192

line 306, 'provide a unique' instead of 'provide with a unique'

Reply: thank you for this observation, correction have been made

Correction: line 224

line 316, 'today' instead of 'at today'

Reply: thank you for this observation. We corrected with “now”

Correction: line 235

line 317,  change 'where' to a more appropriate word

Reply: thank you for this observation. We corrected with the sentece

Correction: line 250

line 343, 'in type...'

Reply: this comment is not clear, please specify which word you would to be corrected

Correction: none

line 354, 'we did not identify' instead of 'we not identified'

Reply: thank you for this observation, correction have been made

Correction: line 275

line 365, change 'assumed' to a more appropriate word

Reply: thank you for this observation, correction have been made

Correction: line 285

line 369, 'chemotherapy' instead of 'chemiotherapy'

Reply: thank you for this observation, correction have been made

Correction: line 289

lines 392, 399, and 426 'almost' is redundant

Reply: thank you for this observation, correction have been made

Correction: line 314, 321, 363

For line 356 (new) “Performing CGP analysis for all thymoma patients is relatively questionable...”

line 406, 'the novelty of the topic analysed in the study is that it is an...'

Reply: thank you for this observation, correction have been made

Correction: line 336

line 409, 'results are opening'

Reply: thank you for this observation, correction have been made

Correction: line 339

Also, there are several instances of inappropriately split words with hyphens (ob-served, thy-momas, be-cause,...) – corrected

Reply: thank you for this observation, all the inappropriate splits have been removed

Correction: through the text

Round 2

Reviewer 1 Report

Comments and Suggestions for Authors

I appreciate the authors improvement of the manuscript after the first round of reviews.

Major comments:

Figure 1 (based on the data in Supplemental Table 1), the sample type labels are wrong for at least patients 1 and 2. For example, Patient 2 only has a variant in the relapse, but this is labeled as primary in the figure. Please review and correct.

Additionally, the version of Figure 1 supplied to the reviewers is basically illegible. This has been additionally noted to the managing editor.

Minor comments:

I still think there are a number of spelling and grammar errors that could be improved.

When describing the NGS failures, it would be clearer to say NGS failure in either primary or recurrence, rather than in one sample.

The first line of data in Table 4 is confusing and at first I thought it was inaccurate. There are only 19 recurrences that were sequenced in this study, so please indicate in the title that these are gene mutations observed in any specimen for patients with recurrent thymomas.

The format of Table 4 makes the data difficult to read with data from the same concept out of alignment (rows are not aligned).

The ID for specimen 24 is incorrect in Supplemental Table 1.

Overall, the additional supplemental tables are of utility to the community and help clarify some of the points being made as well as some of the challenges to the study.

Comments on the Quality of English Language

There are still minor grammar and spelling errors that could be corrected with spell/grammar check.

Author Response

I appreciate the authors improvement of the manuscript after the first round of reviews.

Major comments:

Figure 1 (based on the data in Supplemental Table 1), the sample type labels are wrong for at least patients 1 and 2. For example, Patient 2 only has a variant in the relapse, but this is labeled as primary in the figure. Please review and correct.

Additionally, the version of Figure 1 supplied to the reviewers is basically illegible. This has been additionally noted to the managing editor.

Reply: Thanks for your correction. Figure 1 has been revised according to Your suggestion and a new version with higher quality has been updated (please see pdf file)

Correction: Figure 1

Minor comments:

I still think there are a number of spelling and grammar errors that could be improved.

When describing the NGS failures, it would be clearer to say NGS failure in either primary or recurrence, rather than in one sample.

Reply: Thanks for your correction. We completely agree with your suggestion. Figure 2 has been revised according by including your sentence

Correction: Figure 2

The first line of data in Table 4 is confusing and at first I thought it was inaccurate. There are only 19 recurrences that were sequenced in this study, so please indicate in the title that these are gene mutations observed in any specimen for patients with recurrent thymomas.

The format of Table 4 makes the data difficult to read with data from the same concept out of alignment (rows are not aligned).

Reply: Thanks for your correction. Even in this case, we agree with your revision and we have modified the text and the footnote as reported herein

Correction:

Text:  Finally, by exploring the associations between other clinical variables and gene mutations (identified in any type of specimen in this group, i.e. primary and/or recurrence), we observed a significantly higher frequency of genetic alteration in DNA-repair pathways in early Masaoka-Stage tumors (see Table 4). Similar gene profile distribution was found according to age, presence of M.G., histology and classes of risk.

Footnote  Table 4. Relationship between clinic-pathological variables and gene mutations in Recurrent Thymoma (any specimen, see text).

The ID for specimen 24 is incorrect in Supplemental Table 1.

Reply: Thanks for your correction. Suppl Table 1 was corrected

Correction: Suppl Table 1

Overall, the additional supplemental tables are of utility to the community and help clarify some of the points being made as well as some of the challenges to the study.

Comments on the Quality of English Language

There are still minor grammar and spelling errors that could be corrected with spell/grammar check.

Reply: Thanks for your revisions. The entire manuscript has been reviewed by a B3-English level for correcting errors.

Correction: trough the text